# Spatial population genetic structure of *Caquetaia kraussii* (Steindachner, 1878) evidenced by species-specific microsatellite loci in the middle and low basin of the Cauca River, Colombia

Isaí Cataño Tenorio, Cristhian Danilo Joya, Edna Judith Márquez *

Sede Medellín, Facultad de Ciencias, Escuela de Biociencias, Grupo de Investigación de Biotecnología Animal, Universidad Nacional de Colombia, Medellín, Colombia

* ejmarque@unal.edu.co

## Abstract

The adaptative responses and divergent evolution shown in the environments habited by the Cichlidae family allow to understand different biological properties, including fish genetic diversity and structure studies. In a zone that has been historically submitted to different anthropogenic pressures, this study assessed the genetic diversity and population structure of cichlid *Caquetaia kraussii*, a sedentary species with parental care that has a significant ecological role for its contribution to redistribution and maintenance of sedimentologic processes in its distribution area. This study developed *de novo* 16 highly polymorphic species-specific microsatellite loci that allowed the estimation of the genetic diversity and differentiation in 319 individuals from natural populations in the area influenced by the Ituango hydroelectric project in the Colombian Cauca River. *Caquetaia kraussii* exhibits high genetic diversity levels (Ho: 0.562–0.885; He: 0.583–0.884) in relation to the average neotropical cichlids and a three group-spatial structure: two natural groups upstream and downstream the Nechí River mouth, and one group of individuals with high relatedness degree, possibly independently formed by founder effect in the dam zone. The three genetic groups show recent bottlenecks, but only the two natural groups have effective population size that suggest their long-term permanence. The information generated is relevant not only for management programs and species conservation purposes, but also for broadening the available knowledge on the factors influencing neotropical cichlids population genetics.

## Introduction

Cichlids are one of the most diverse families, having 1756 valid species distributed in 257 genera, and having a notable presence in Africa, and Central and South America [1]. Their remarkable divergent evolution and fast adaptation to different environments make them an evolutionary model for researching their biological properties [2–6]. Their adaptive radiation

**Data Availability Statement:** All relevant data are within the manuscript and its Supporting information files.

**Funding:** This study was funded by Universidad Nacional de Colombia Sede Medellín and Empresas Públicas de Medellín, Grant CT-2019-000661 "Variabilidad genética de un banco de peces de los sectores medio y bajo del Río Cauca". Funders have not play any role in the study design, data collection and analysis, decision to publish, or preparation of the manuscript.

**Competing interests:** The authors have declared that no competing interests exist.

may be the result of the combination of intrinsic factors and the environment, namely, the interaction between morphological or behavioral innovations and ecological opportunities [6–9]. Furthermore, plasticity in genetic, morphological, and reproductive features allow this species to survive in different environments, take advantage of ecological opportunities and, ultimately, to diversify [10–13]. Another important factor is hybridization, which has provided the species with advantageous genetic variability to face the environmental changes, differentiate or remain in their surroundings [6, 14, 15].

Most cichlids have shown particular life histories such as parental care [16], assortative mating [17–21] and a tendency to sedentarism [22, 23], features that influence the genetic diversity patterns of various sea and freshwater fish [24]. Namely, fish species that reach maturity at an early age show higher levels of genetic variability than the species that reach maturity later [24]. Besides, life expectancy is inversely correlated to genetic diversity as individuals with lower life expectancy often have higher genetic diversity [25, 26]. Moreover, individuals that migrate and reproduce with distant populations exhibit gene flow and, thus, high genetic diversity [24, 25].

On the other hand, sedentary species and/or species having parental care with egg or larvae retention display little to no gene flow which may lead to genetic isolation among populations if the effective population size is not large enough [27]. As for other species, habitat-preference population isolation in cichlids may be influenced by environmental heterozygosity or geographical barriers [28]. Interestingly, some cichlids avoid reproducing with close relatives spreading to habitats occupied by unknown individuals, which allows preventing inbreeding and keeping the genetic diversity [29].

In neotropical cichlids, studies of population genetics on natural populations have focused on sedentary species such as *Geophagus brasiliensis* [22], *Cichla temensis* [23], *Apistogramma agassizii* [30], *Geophagus aff. Brasiliensis* [31], *Apistogramma gephyra* and *Apistrogramma pertensis* [32], *Pterophyllum scalarae* [33] and *Cichla ocellaris* var. *kelberi* [34], that exhibit moderated genetic diversity and spatial genetic structured populations. Contrastingly, rheophile species *Gymnogeophagus setequedas* exhibited high genetic diversity and absence of spatial genetic structure when analyzed with the heterologous microsatellite loci developed for *Geophagus brasiliensis* [35]. Additionally, genetic diversity and inbreeding were tested using heterologous microsatellites for *Oreochromis niloticus*, a species kept captive for food production [36].

To contribute to the knowledge of neotropical cichlids population genetics, this study selected yellow 'mojarra' *Caquetaia kraussii* (Steindachner, 1878), one of the 104 cichlid species described for Colombia and one of the four species registered in the Magdalena-Cauca basin [37]. This species, also naturally distributed in the basins of rivers Atrato, Sinú and Maracaibo, is subject to artisanal fishery for human consumption, performs important ecosystem services like redistribution and maintenance of sedimentologic processes [38], and preferentially habits swamps, ponds, and undisturbed waters in the lower areas of the rivers and streams in altitudes of up to 500 m s. n. m. [39]. Regarding reproductive features, *C. kraussii* shows an average generational time of eight months [40], has partial spawning events through the year in the Atrato [41] and Sinú [42] rivers, an equilibrium reproductive strategy associated with low relative fecundity, parental care, and sedentary tendency [42, 43].

Furthermore, *C. kraussii* is in the Ituango hydroelectric project (PHI, for its acronym in Spanish, from Bolombolo of Venecia, Antioquia to Pinillos, Bolívar) influence area. This project encompasses the last 500 km of the Cauca River, which is 1350 km long and has a 63,300 km² area [44]. Some studies have noted that equilibrium strategy species like *C. kraussii* are threatened for their habitat fragmentation due to hydroelectric constructions [45]; nonetheless, the effects of these anthropogenic activities may vary since downstream there is a

reduction in the lateral and longitudinal conductivity and the flow regime stabilizes while the conditions in the dam change from lotic to lentic, increasing the thermal stratification [46]. Historically, this area has been exposed to the negative influence of other anthropogenic factors such as water contamination, livestock, fishing, and mining [47], that may explain the genetic bottlenecks in various species sampled prior to the hydroelectric construction [48–50].

The impact of hydroelectric plants on cichlids in neotropical environments is poorly documented and fish responses appear to depend on their life history [35], the duration of fragmentation [31], and potential impacts resulting from hybridization between invasive or introduced species [34]. For example, *Gymnogeophagus setequedas*, an endangered cichlid that seems to prefer fast waters, was reported to have disappeared after the construction of the Itaipu reservoir [51]. However, it exhibited gene flow in lotic environments of the Iguaçu River basin [35]. For populations fragmented for approximately 17 years or longer, *Geophagus* aff. *brasiliensis* and other neotropical non-migratory fish species show significant genetic structure [31]. Recent evidence of hybridization between two introduced species in a reservoir (*Cichla ocellaris* var. *kelberi* and *Cichla piquiti*) and the high genetic diversity found using microsatellite loci raises concerns about its indication of a possible increase in local adaptability that could enhance establishment success in novel areas [34].

Namely, this study aimed to provide a response to *C. kraussii* genetic diversity and structure related questions in the medium and lower sections of the Cauca River. Since *C. kraussii* is in strongly anthropogenic pressured habitats and is a sedentary species with parental care, the *a priori* expectation was for it to exhibit low genetic diversity and spatial structure in the PHI influence area. To contrast these hypotheses, microsatellite loci were developed as a molecular tool to assess this species genetic diversity and structure. This approach was used for avoiding heterologous loci-related genotyping errors [52–55], considering that to date microsatellite loci have been developed for phylogenetically distant neotropical cichlids like *Amphilophus cichlasoma* [56], *Symphysodon discus* [57], *Astronotus crassipinis* [58], *Cichla piquiti* [59], *Cichla monoculus* [60], *Cichla temensis and Cichla orinocensis* [61], *Geophagus brasiliensis* [62] *Apistogramma agassizii* [63], *Apistogramma gephyra* [64] and *Pterophyllum scalarae* [33].

## Materials and methods

### Study area

This study assessed fin and muscle tissues preserved in ethanol 95%, obtained from 319 *C. kraussii* individuals captured in different sections of the Cauca River between 2020 and 2022 (Fig 1) by Universidad de Antioquia, Universidad de Córdoba, and Universidad Nacional de Colombia Sede Medellín, through scientific cooperation agreement CT-2019-000661, under environmental license # 0155 of January 30th, 2009, from Ministry of Environment, Housing and Territorial Development for the Ituango hydroelectric construction. These sections previously identified [65] include the medium (PHI: 100 samples) and lower (S4, S5, S6, S7 and S8: 21, 43, 80, 49 and 26 samples, respectively) sections of the Cauca River. The PHI section, which is 46 km long, corresponds to the dam zone, a lentic system that before the hydroelectric construction had rapids and strong streams. The remain sections (S4: 38 km, S5: 61 km, S6: 78 km, S7: 29 km, S8: 17 km) are downstream the dam and comprise lentic (swamps) and lotic (streams and rivers) systems in a floodplain influenced by the Nechí River mouth.

### Microsatellite loci design

Microsatellite loci were designed following Landínez-García & Márquez [66]. To this end, genomic DNA (gDNA) was extracted from a *C. kraussii* individual with QIAamp DNA Mini Kit (Qiagen). A genomic library was created from said DNA using Truseq Nano DNA Library

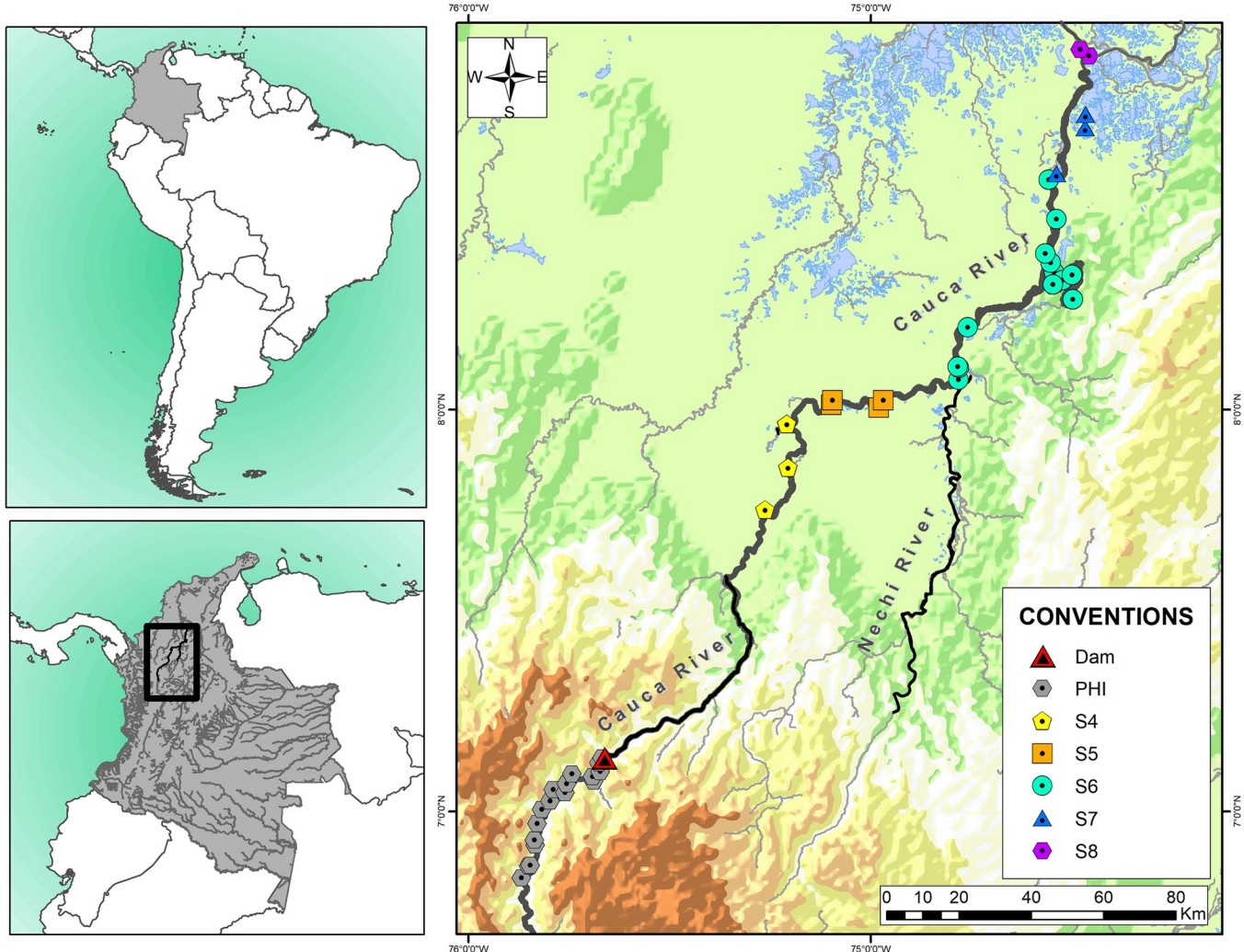

**Fig 1.** *Caquetaia kraussii* **sampled sites in the Cauca River, Colombia.** PHI: Ituango hydroelectric project, 17 sites (purple). S4: 3 sites (light blue). S5: 5 sites (blue). S6: 13 sites (red). S7: 3 sites (orange). S8: 2 sites (yellow). Self-made creation of the map based on contour lines scaled 1:100,000 from the Instituto Geográfico Agustín Codazzi source, 2019 (Available from: IGAC Geoportal, https://geoportal.igac.gov.co/contenido/datos-abiertos-cartografia-y-geografia).

Prep Kit, and sequencing was performed through Illumina MiSeq, generating 300 base paired end reads. Then, raw read cleaning was carried out with CUTADAPT v2.10 [67], sequences containing the microsatellites were selected with PAL_FINDER v0.02.03 [68], and PRIMER3 v2.0 [69] was used for the primer designs in the flanking sequences of the microsatellite. Subsequently, the correct evaluation of the primers was tested *in silico* with electronic PCR [70].

To test polymorphism in 30 selected loci, DNA was extracted from 24 individuals of all of the sections using GF-1 kit (Vivantis). PCR reactions were carried out in final volumes of 10 µl, using 0.22 X Master Mix (Invitrogen) buffer, 2.5% v/v enhancer, 0.25 pmol/µl forward primer, 0.5 pmol/µl reverse primer (Macrogen), 0.25 pmol/µl adapters [71] fluorescently labeled (Applied Biosystems FAM, VIC, NED, or PET) at the 5' end and ~30 ng/µl DNA. Thermal profile, carried out in a thermocycler T100 (Bio-Rad, CA, USA), comprised a heating step at 95 ˚C for 3 m, followed by 40 denaturalization cycles at 90 ˚C for 30 s and an annealing step at 50 ˚C for 40 s, with no final elongation. Amplicons were separated by capillary

electrophoresis using SeqStudio (Applied Biosystems) with GS600 LIZ as molecular weight marker. Finally, GeneMarker v3.0.0 (SoftGenetics LLC [R]) carried out the genotypes reading and recording, selecting 16 species-specific microsatellite loci according to their polymorphism level.

## Population genetics analysis

Prior to assessing *C. kraussii* genetic diversity, ARLEQUIN v3.5.2.2 [72] was used for evaluating expected (He) and observed (Ho) heterozygosities, inbreeding coefficient ($F_{IS}$), and departures from Hardy-Weinberg equilibrium and linkage disequilibrium among loci, applying, in the latter, the sequential Bonferroni correction for the statistical significance in the multiple comparisons [73]. BayeScan v2.1 [74] was used for identifying outlier loci presence. The number of alleles per locus (Na) and the allelic range (Ra) were calculated with GenAlex v6.51 [75] and the polymorphism information content (PIC) was determined using Cervus v3.0.7 [76]. GENEPOP 4.7 [77, 78] was used for finding the multilocus values (*across-loci*) and FSTAT v 2.9.4 [79] for the allelic richness (Ar). The correlation analysis between He, Ar, and estimators of genetic differentiation (described below) with the physical (km) or geographical distances of the sections of the Cauca River mouth in the Magdalena River was performed using Pearson correlation with the R-packages ecodist and GGPLOT2 [80, 81].

The relatedness coefficient (rxy) estimation among all possible pair of individuals was analyzed using the R-related package [82]. In absence of a pedigree, relatedness inferences for non-model species were based in simulated relatedness measures from empirical data [83]. Initially, the best estimator between maximum likelihood (Dyadml in [84] and Trioml in [85]) and non-maximum likelihood (Lynchli in [86]; Lynchrd in [87]; Quellergt in [88]; Wang in [89]) estimators was determined from simulated relatedness values among 100 pairs. The best estimator for analyzing the data was the one having relatedness estimations with the highest correlation between the simulated data from the empirical allelic frequencies and the theorical values, in each of the four relatedness categories: unrelated (UR, rxy = 0.00), half sibs (HS, rxy = 0.25), full sibs (rxy = 0.50) and parent–offspring (PO, rxy = 0.50). Once the best estimator was selected for each species, pairwise relatedness was estimated. The same estimator was used for calculating individual inbreeding with COANCESTRY [90].

Genetic structure was determined through the analysis of molecular variance (AMOVA) and calculation of estimators $F'_{ST}$ [91, 92] and Jost's $D_{EST}$ [92, 93] using GenAlEx v6.51b2 [75]. The Bonferroni correction [73] was applied for the statistical significance of the estimators. Furthermore, a discriminant analysis of principal components (DAPC) was performed using the R-package Adegenet [94], with 32 principal components (a-score: 0.422; explaining 60% of total variation) and six discriminant functions retained. Other approach included a Bayesian analysis in STRUCTURE 2.3.4 [95], with 1,000,000 Monte Carlo Markov chains with 100,000 regarded as the burn-in period, admixture model and correlated allele frequencies as a priori models. This analysis was repeated 20 times for each K, assuming 1 to 10 K. Then, StructureSelector [96] was used for determining the best estimation of K value based on Puechmaille [97] estimators MEDMEANK, MEDMEDK, MAXMEANK and MAXMEDK, ΔK [98] and Ln Pr (XlK) [99], and the integrated software Clumpak [100] was utilized for graphically representing the results. Individuals were assigned to their respective genetic stocks in accordance with the co-ancestry estimators and they were submitted to population genetics analysis following the above-mentioned methodology.

Recent bottleneck detection was performed through two approaches: Bottleneck v.1.2.02 [101] for calculating excess heterozygosity ($H_E > H_O$ mutational equilibrium assumption) under the infinite alleles (IAM), stepwise mutational (SMM), and two-phase (TPM; parameter

settings: IAM: 10%, SMM: 90%, Variance: 10.00, Probability: 90%) models, through Wilcoxon signed rank test with 1000 iterations [102], and ARLEQUIN v3.5.2.2 [72] for calculating the M Ratio [103]. Moreover, the population effective size was estimated with the linkage disequilibrium method [104] implemented in NeEstimator v2.1 [105] considering the allelic frequency of 0.05. Following Lonsigner and company [106], all individuals were included in the Ne analysis, even those with some relatedness degree, since including related individuals generates little bias in the linkage disequilibrium method [107] while small population sizes may cause large bias in the estimation [108].

Lastly, GENECLASS2 [109] was used for testing whether an individual resides in the sampled site or is immigrant from another section. Analysis was carried out with the Bayesian method [110], and simulations with the Monte Carlo resampling method [111] with 16 loci, 10,000 and 0.01 as threshold of type I error. Likewise, the BA3MSAT extension of the BAYE-SASS software [112, 113] was employed for the recent estimation of gene flow between sections, utilizing 50,000,000 MCMC, with a burn-in of 5,000,000 and sampling intervals of 5,000. Delta values for migration rates (deltaM), allele frequencies (deltaA), and inbreeding coefficients (deltaF) were set at 0.49, 0.39, and 0.46, respectively. Convergence was assessed using the Tracer v1.7.2 program [114].

## Results

### Microsatellites development and detection of outlier loci

A total of 16 (4mer: 15; 5mer: 1) out of 30 preselected loci satisfied the polymorphic criteria (Table 1). The remaining loci either exhibited inconsistent amplification, low levels of polymorphism, or were monomorphic (S1 Table). In the 319 samples (S2 Table), the allelic size range oscillated between 110–369 pb, the Na between 9–29 (Ckra13/Ckra 27; Ckra24) and PIC between 0.69–0.915 (Ckra 13; Ckra01). Moreover, all loci showed linkage equilibrium, indicating independent segregation (S3 Table), and Hardy-Weinberg equilibrium in most of the sampled sites (S4 Table), indicating that departures are not attributable to technical causes.

Furthermore, Bayescan posterior probability (PO) values for the loci detection under selection (S5 Table) evidenced that the only paired comparison showing one locus under selection (Ckra 21; reference value > 0.76) was S4-S5 v. S6-S7-S8. Moreover, another parameter found, Alpha, had a value of -1.234 which suggests the existence of balancing or purifying selection. The results presented below are based on 16 loci, as consistent findings were obtained when including or excluding this locus in subsequent analyses.

### Population genetic diversity

Genetic diversity by site showed a slight heterozygosity deficit in S4 and two genetic groups (S6-S7-S8, S4-S5) revealed by the genetic structure analysis (Table 2). This deficit was related with a significant inbreeding only in S4 ($F_{IS}$: 0.058; $P_{FIS}$: 0.036). Genetic diversity showed a decreasing gradient from the lower to the middle section of the Cauca River (S8>S7>S6>S5>S4; Table 2; S1 Fig) and reached its lower values in the confined environment PHI. This gradient remained still when diversity was compared among genetic groups (S6-S7-S8>S4-S5>PHI). The distribution of genetic diversity was negatively related with the distance to the Cauca River mouth (S1 Fig), for both He (R: -0.970, p: 0.001) and Ar (R: -0.980, p < 0.0001).

The relatedness coefficient using estimator Dyadml that showed the higher correlation (Dyadml = 0.915, Trioml = 0.913, Wang = 0.912, Linchli = 0.909, Quellergt = 0.905, Lynchrd = 0.826; S6 Table) indicates a high percentage of unrelated pairs of *C. kraussii* in the global analysis (86%), stocks (S4-S5: 76%; S6-S7-S8: 88%) and sections where the species is

**Table 1. Primer sequences, features, and genetic diversity of the 16 *Caquetaia kraussii* species-specific microsatellite loci.**

| Locus | Motif | F (forward) and R (reverse) primers | Ra | Na | PIC | Adapter |
|---|---|---|---|---|---|---|
| Ckra01 | (ATCT)n | F: CATGCAGTTATCACATTATTGTCC<br>R: CATCACGTAGTATGGCACTCC | 205–337 | 28 | 0.915 | Tail B |
| Ckra02 | (ATGG)n | F: AGGCCAAAAGATGGATGGG<br>R: TTGAACAAAATACCTTAGCCTCAGC | 219–291 | 17 | 0.778 | Tail B |
| Ckra03 | (ATCT)n | F: CCAGAACAAAATGCTCACTGC<br>R: GTGGCCAATAAAACATAAAGACC | 110–170 | 15 | 0.843 | Tail B |
| Ckra04 | (ATCT)n | F: CAATAGCCTACACTCTGGACAGG<br>R: GCCTGTCGGTCAAATGTAGC | 128–228 | 24 | 0.910 | Tail B |
| Ckra05 | (ATCT)n | F: GGATGCTCATATTGAGCGTAACC<br>R: GTTCGAAGTATCCTTGGGCG | 274–354 | 20 | 0.882 | Tail B |
| Ckra06 | (ATGG)n | F: TCGCTTCATAGAAATGTTGTTGG<br>R: TCTGTTGAGTCTGTTGGGGC | 163–215 | 14 | 0.785 | Tail B |
| Ckra07 | (ATCT)n | F: ACACATGTCAGGTGGATGGG<br>R: GTCACTGACTCTGCATACCAGC | 207–319 | 25 | 0.910 | Tail B |
| Ckra08 | (ATGG)n | F: AACATCCTGCAGCATTCACG<br>R: TGACCCTGAAAAGGATACATGG | 166–206 | 11 | 0.798 | Tail B |
| Ckra12 | (ATGG)n | F: ATGATGTGCTGATGGATGGG<br>R: CGCCAATGAATTGGATAAGTGG | 255–299 | 12 | 0.768 | Tail A |
| Ckra13 | (ATGG)n | F: AGACCCTGAACAGGATAAGTGG<br>R: GAGGCTGACCAGAGGAAAGG | 222–254 | 9 | 0.690 | Tail A |
| Ckra18 | (ATGG)n | F: TGAAACAAACTGGTTGGAAGG<br>R: ATAACCCAAAACAGGGCACC | 160–220 | 16 | 0.790 | Tail D |
| Ckra21 | (ATCT)n | F: GTGGAGACGACACCAAGTGC<br>R: TGGCTTATGGATGAAGCACC | 232–352 | 23 | 0.864 | Tail D |
| Ckra22 | (ATGG)n | F: ACATGGAGCTGATTCCAGCC<br>R: AGGTGACTTCGCCTCTCACC | 210–282 | 19 | 0.881 | Tail D |
| Ckra24 | (ATCT)n | F: CACCCTGTTGTGGTTAACGG<br>R: GAATAATGCAGCAGCAAGGC | 241–369 | 29 | 0.927 | Tail D |
| Ckra27 | (ATGG)n | F: CTGTGGCAGCTGGGATAAGC<br>R: AGGGTTCCTGCAAACACAGG | 156–188 | 9 | 0.731 | Tail C |
| Ckra29 | (ATATC)n | F: TCCAAACACGGTCAGTCTGC<br>R: AGTGGGCCTATTGTTGGGG | 185–240 | 12 | 0.838 | Tail C |

Ra: Allelic range. Na: number of alleles per locus. PIC: polymorphism information content. Tail A: GCCTCCCTCGCGCCA. Tail B: GCCTTGCCAGCCCGC. Tail C: CAGGACCAGGCTACCGTG. Tail D: CGGAGAGCCGAGAGGTG.

**Table 2. *Caquetaia kraussii* genetic diversity in the middle and lower sections of the Cauca River, Colombia.**

| Section | N | Na | Ar | Ho | He | $P_{HWE}$ | $F_{IS}$ | $P_{FIS}$ |
|---|---|---|---|---|---|---|---|---|
| S8 | 26 | 14.063 ± 4.683 | 12.845 ± 4.063 | 0.874 ± 0.080 | 0.884 ± 0.069 | 0.598 | 0.007 | 0.360 |
| S7 | 49 | 14.813 ± 5.180 | 11.977 ± 3.787 | 0.885 ± 0.067 | 0.880 ± 0.058 | 0.295 | -0.008 | 0.732 |
| S6 | 80 | 14.938 ± 5.285 | 11.178 ± 3.423 | 0.862 ± 0.083 | 0.871 ± 0.061 | 0.164 | 0.008 | 0.249 |
| S5 | 43 | 11.375 ± 3.810 | 9.501 ± 2.651 | 0.835 ± 0.084 | 0.823 ± 0.080 | 0.143 | -0.020 | 0.905 |
| S4 | 21 | 9.688 ± 3.572 | 9.424 ± 3.469 | 0.727 ± 0.130 | 0.779 ± 0.115 | **<0.0001** | 0.058 | **0.036** |
| Stock S6-S7-S8 | 155 | 17.125 ± 6.076 | 15.255 ± 5.393 | 0.871 ± 0.070 | 0.879 ± 0.059 | **0.046** | 0.006 | 0.232 |
| Stock S4-S5 | 64 | 12.438 ± 4.427 | 12.362 ± 4.391 | 0.800 ± 0.089 | 0.812 ± 0.092 | **0.006** | 0.008 | 0.289 |
| PHI | 100 | 7.250 ± 2.017 | 5.475 ± 1.225 | 0.664 ± 0.128 | 0.667 ± 0.121 | 0.128 | 0.001 | 0.493 |

N: number of individuals. Na: number of alleles per locus. Ar: allelic richness. Ho: observed heterozygosity. He: expected heterozygosity. $P_{HWE}$: p-value for the departure from Hardy-Weinberg equilibrium. $F_{IS}$: inbreeding coefficient. $P_{FIS}$: p-value for the inbreeding. Values in bold denote statistical significance.

**Table 3.** *Caquetaia kraussii* pairwise comparisons of the standardized statistics $F'_{ST}$ and $D_{EST}$ in lower and middle sections of the Cauca River, Colombia.

|  | PHI | S4 | S5 | S6 | S7 | S8 |
|---|---|---|---|---|---|---|
| $F'_{ST}$ pairwise comparisons | | | | | | |
| PHI | - | 0.001 | 0.001 | 0.001 | 0.001 | 0.001 |
| S4 | **0.108** | - | 0.005 | 0.001 | 0.001 | 0.001 |
| S5 | **0.080** | **0.013** | - | 0.001 | 0.001 | 0.001 |
| S6 | **0.085** | **0.035** | **0.022** | - | 0.001 | 0.002 |
| S7 | **0.079** | **0.038** | **0.024** | **0.007** | - | 0.254 |
| S8 | **0.079** | **0.040** | **0.028** | **0.011** | 0.008 | - |
| $D_{EST}$ pairwise comparisons | | | | | | |
| PHI | - | 0.001 | 0.001 | 0.001 | 0.001 | 0.001 |
| S4 | **0.584** | - | 0.007 | 0.001 | 0.001 | 0.001 |
| S5 | **0.478** | **0.033** | - | 0.001 | 0.001 | 0.001 |
| S6 | **0.597** | **0.265** | **0.197** | - | 0.001 | 0.002 |
| S7 | **0.553** | **0.289** | **0.218** | **0.036** | - | 0.267 |
| S8 | **0.545** | **0.292** | **0.238** | **0.062** | 0.008 | - |

Below diagonal, estimator value found. Upper diagonal, significant statistic. Values in bold denote statistical significance.

naturally distributed (S4: 67%; S5: 77%; S6: 84%; S7: 88%; S8: 92%), with a low number of individuals which individual inbreeding exceeds 10% (S4: 8; S5: 9; S6: 15; S7: 6; S8: 3). The contrary was observed in sections with explicit anthropic intervention that exhibited low percentages of unrelated pairs (PHI: 8.4%) and high percentages of pairs with some relatedness degree (PHI: 91.6%), with a high number of individuals which individual inbreeding exceeds 10% (PHI: 78).

## Population genetic differentiation

AMOVA ($F_{ST}$ = 0.935, P value = 0.001) and pairwise comparisons of the standardized statistics $F'_{ST}$ and $D_{EST}$ showed significant differences among sites (Table 3), evidencing that all sections are genetically different from each other, except for S7 and S8 in both tests.

Both the DAPC (Fig 2) and the Structure population assignment Bayesian analysis (Fig 3) suggested that *C. kraussii* is formed by three or four genetic groups ($\Delta K$ = 2; MedMed = 3; MedMean, MaxMed, MaxMean = 4; Mean LnP(K) = 3). Ultimately, three stocks were determined: PHI, stock S4-S5 and stock S6-S7-S8.

## Population genetic demography

Bottleneck tests summarized in Table 4 were significant in all sites (except S4) under the IAM; S6 and S7 and genetic group S6-S7-S8 under the TPM, and none of the evaluated groups was significant under the SMM. Additionally, M ratio was below 0.680 indicating recent bottlenecks in all sites and genetic groups. Moreover, effective population sizes were superior to 1,000 in S7 (1665) and stocks S4-S5 (2220), S6-S7-S8 (5349), exhibited low values in PHI (161), S4 (49) and S6 (723) and it was not possible to precisely estimate said sizes for S5 and S8 (Table 4). As for migratory events, eight individuals were detected as potentially immigrants in the sampled sites and were assigned to the most likely origin sections; a S4 individual in S5 (p value: 0.005); two S5 individuals in S4 (p value: 0.007); a S6 individual in S7 (p value < 0.0001); and a S7 individual in S6 (p value: 0.002). Results from BayesAss (S7 Table) showed that the highest migration rates were observed within each section, ranging from 67.3% to 98.1% (m

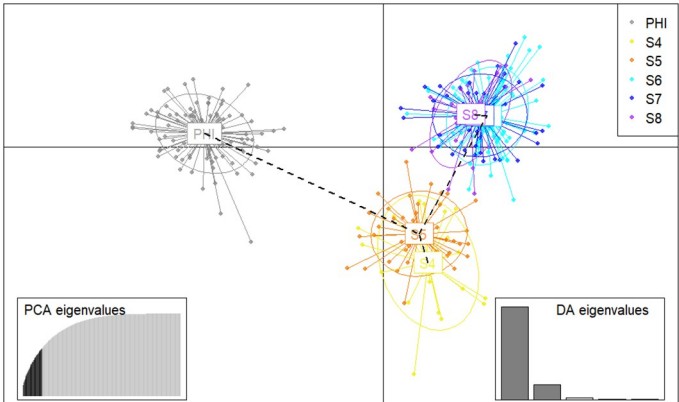

**Fig 2. Discriminant analysis of principal components of *Caquetaia kraussii* of seven sections in the Cauca River, Colombia.**

[PHI][PHI]: 0.981±0.008, m[S6][S6]: 0.963±0.013, m[S5][S5]: 0.940±0.021, m[S4][S4]: 0.679 ±0.012, m[S8][S8]: 0.677±0.010, m[S7][S7]: 0.673±0.006). Additionally, a higher migration rate upstream (25.8%) was observed between S4 and S5, while higher migration rates downstream (26.2%–29.5%) were observed between S6, S7, and S8. The remaining migration rates ranged from 0.3% to 1.9%.

Mantel test showed correlation between the distance matrixes with 0.1 as threshold of the type I error, where genetic distances were significantly correlated with the geographical distances ($R_{F'ST}$ = 0.633, p = 0.045; $R_{DEST}$ = 0.717, p = 0.009) (S2 Fig).

## Discussion

This study demonstrated the presence of spatial structure and high levels of genetic diversity of *C. kraussii* populations in the middle and lower sections of the Cauca River. Additionally, this study provides a group of 16 *C. kraussii* species-specific microsatellite loci with long repetition motifs (4-mer, 5-mer) and are highly polymorphic since its PIC values > 0.5 were above the range proposed by Botstein and company [115]. Both the diversity levels and the PIC values of the loci here developed were found within the range of the loci designed for other neotropical cichlids which repetition motifs are mostly short (2-mer) and compound [33, 56–64]. Additionally, loci here developed exhibited amplification consistency and high definition of the electropherograms that ease the allele assignation and, thus, their genotypification, for which they are considered informative and appropriate for the *C. kraussii* population genetics study.

According to the *a priori* expectations, *C. kraussii* showed a spatial genetic structure that may result from features of its life history for being a resident species with parental care [40, 43], which limits the gene flow and increases the differentiation of its genetic pool. A similar

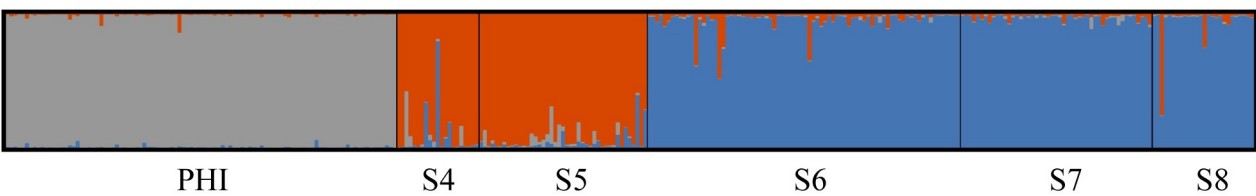

**Fig 3. Graph bar of the population co-ancestry coefficient estimated by Structure for K = 3.**

**Table 4. Recent bottlenecks detection tests and effective population size in *Caquetaia kraussii* populations.**

| Sections and stocks | IAM | SMM | TPM | M Ratio | Ne |
|---|---|---|---|---|---|
| S8 | <0.0001 | 0.628 | 0.248 | 0.234 ± 0.030 | Infinite (460, Infinite) |
| S7 | <0.0001 | 0.628 | 0.017 | 0.229 ± 0.030 | 1665 (350, Infinite) |
| S6 | <0.0001 | 0.684 | 0.042 | 0.240± 0.029 | 723 (360, 15881) |
| S5 | <0.0001 | 0.989 | 0.768 | 0.210 ± 0.037 | Infinite (454, Infinite) |
| S4 | 0. 0800 | 1.000 | 1.000 | 0.219 ± 0.033 | 49 (31, 101) |
| S6-S7-S8 | <0.0001 | 0.851 | 0.025 | 0.240 ± 0.029 | 5349 (1157, Infinite) |
| S4-S5 | <0.0001 | 1.000 | 0.983 | 0.225 ± 0.036 | 2220 (364, Infinite) |
| PHI | 0.0040 | 0.999 | 0.942 | 0.180 ± 0.042 | 161 (114, 254) |

IAM: infinite alleles model. SMM: stepwise mutational model. TPM: two-phase model. Probabilities according to Wilcoxon signed rank test (excess heterozygosity), calculated through Bottleneck v1. M ratio values calculated by Arlequin v3.5.2.2. Ne: number of individuals of the effective population size, calculated by NeEstimator v2.1. Values in bold denote statistical significance.

result was found for other cichlid species like *Cichla temensis* [23], *Geophagus brasiliensis* [22], *Apistogramma agassizii* [30], *Geophagus aff. Brasiliensis* [31], *Apistogramma gephyra* and *Apistrogramma pertensis* [32], *Pterophyllum scalarae* [33] and *Cichla ocellaris* var. *kelberi* [34].

Spatial structure due to events such as interruption of migratory pathways are not expected for sedentary fish species, as they typically exhibit a spatial genetic structure associated with their life history. In such cases, the potential threats associated with hydroelectric constructions may be related to factors that affect fish species downstream, such as fluctuations in nutrient transportation and shifts in oxygen concentrations [116], water quality and temperature, sediment accumulation, variations in water flow, alterations in the morphology of the main river channel and swamps [117–120]. The impact of these factors on the population genetics of *C. kraussii* remains unknown due to the lack of previous information on these subjects. However, the significant gene flow observed in various migratory fish species coexisting in this area [48–50, 121–123] suggests the absence of physical or chemical barriers limiting gene flow in these species.

Because this species is considered resident, a reduction in the genetic similarity among populations is expected as the geographical distance among them increases. In this study, the genetic differentiation estimators were spatially autocorrelated in distances 2–6 times longer than the dispersion range estimated for the species, providing a strong support to the isolation by distance explanation.

Moreover, the analysis for detecting immigrants indicated that PHI show low migration rate from downstream, corroborating the restriction of gene flow among these sections. Due to the rapids and high velocities of waterflow in the zone before the hydroelectric construction, results found in this study on the genetic structure and immigration, along with the low levels of genetic diversity and Ne < 1,000, suggest that *C. kraussii* individuals sampled in PHI may represent a founder effect from populations that do not originate from downstream the dam. Origin of the founder population remains to be explored in future studies. Since 91.6% of the population in this section is formed by individuals that show some relatedness degree, the latter explanation seems more likely than the alternative of a fast genetic differentiation in the short-term between PHI and the remaining populations downstream, caused by new environmental conditions and habitat preferences of *C. kraussii*, despite cichlids appear to respond rapidly to changing conditions.

Under this scenario, *C. kraussii* is formed by two natural stocks, S4-S5 and S6-S7-S8, that are approximately 15 km apart and separated from each other by the Nechí River mouth. The

presence of immigrants among sections was displayed within each stock, while low migration rate between stocks was found. The low dispersion potential of the species and its larvae because of parental care, the variation in its dispersion behavior (upstream in the S4-S5 stock and downstream in the S6-S7-S8 stock), and the confluence magnitude of the Nechí River and the Cauca River may explain the existence of two *C. kraussii* stocks, one upstream and other downstream of the Nechí River mouth. It has been indicated that the confluence position, in addition to the river branching degree, and the asymmetric migration levels downstream influence the genetic variation patterns in the riverside populations showing an increase of 20 times the global genetic diversity in the very branched rivers and of 7 times the genetic differentiation among local populations [124].

Based on its distribution in habitats exposed to anthropogenic activities and their particular life history, the *a priori* expectation was that *C. kraussii* exhibited low genetic diversity in the studied area. In contrast with this *a priori* expectation, Ho (0.562–0.885) and He (0.583–0.884) levels in *C. kraussii* exceeded the heterozygosity levels described for other neotropical cichlids like *Geophagus brasiliensis* (Ho: 0.474–0.628, He:0.534–0.706, [22]), *Cichla temensis* (Ho: 0.183–0.619, He:0.292–0.657, [23]), *Gymnogeophagus setequedas* (Ho:0.593, He: 0.673, [35]), *Apistogramma agassizii* (Ho: 0.364–0.762, He: 0.350–0.754, [30]), *Geophagus aff. brasiliensis* (Ho: 0.532–0.556, He: 0.635–0.640, [31]), *Apistogramma gephyra* (Ho: 0.631–0.662, He: 0.633–0.669) and *Apistrogramma pertensis* (Ho: 0.664–0.742, He: 0.612–0.663; with heterologous loci of *Apistogramma gephyra*) [32], *Pterophyllum scalarae* (Ho: 0.376–0.562, He: 0.512–0.568, [33]) and *Cichla ocellaris* var. *kelberi* (Ho: 0.538–0.733, He: 0.521–0.642, [34]). It is notable that the high diversity of *C. kraussii* was evidenced based on longer repetition motifs (4-mer, 5-mer), which were expected to show less variability than hypervariable short repetitions (2-mer, [125]) mostly used for other neotropical fishes like *Cichla temensis* [61], *Geophagus brasiliensis* [62], *Apistogramma gephyra* [64] and *Pterophyllum scalarae* [33].

Differences in the genetic diversity between *C. kraussii* and the other neotropical cichlids may be caused by typical features of their life history [126], since species that reach maturity earlier or have lower life expectancy show higher levels of genetic diversity [24, 27, 126]. Alternatively, discrepancies may be related to differences in the effective population size, estimator that has been associated with genetic diversity [26]. However, both explanations are difficult to contrast due to the limited information available for neotropical cichlids reproductive features and effective population size; for this reason, it would be convenient to advance in complete studies on its reproduction, population size, and genetic diversity to further explore the factors influencing its population genetics.

Furthermore, genetic diversity measures, He and Ar were negatively related to the distance from the Cauca River mouth to the Magdalena River since genetic diversity decreases as distance to the mouth increases. This genetic diversity distribution pattern of populations is possibly due to habitats diversity downstream the Nechí River mouth, which are preferred by *C. kraussii* and the existing gene flow in this zone, allowing the allelic exchange and maintenance of the high genetic diversity. This idea aligns with genetic data simulation in different landscapes that show that the dendritic net and the riverbed connectivity interfere in the genetic variation distribution [124, 127, 128]. Likewise, the slope of the riverbed may also influence in the genetic variation distribution of a species [129] as in low altitudes there is more diversity of habitats ideal for *C. kraussii* while in medium and lower altitudes the habitats are more limited.

Although the averages of heterozygosity in *C. kraussii* were high, small but significant deficits of Ho were observed in stocks S4-S5 and S6-S7-S8. Since this study used species-specific microsatellite loci, individuals were organized by stocks and the $F_{IS}$ values were not significant, results do not appear to be explained, respectively, by the presence of null alleles, Wahlund

effect, or inbreeding. Another likely explanation would be assortative mating, a behavior observed in other neotropical cichlids like in genera *Geophagus* [19] and *Cichla* [21]. Nonetheless, it is important to remark that there are no available data on this behavior in *C. kraussii*, hence, further exploration on the reproductive behavior of this species is necessary for determining its potential role on the Ho deficit.

Detection of recent bottlenecks in *C. kraussii* in the influence area of the Ituango project matches those found in the same area in *Prochilodus magdalenae* [48], *Pseudoplatystoma magdaleniatum* [49], *Ageneiosus pardalis*, *Pimelodus grosskopfii* and *Sorubim cuspicaudus* [50], which were attributed to anthropic pressures like fishing, mining, and water contamination. These pressures may be the cause of recent bottlenecks in *C. kraussii*, although in this case there is an additional factor related to the habitat alteration because of the PHI dam construction. Additionally, recent bottlenecks have also been detected in other cichlid species as in *Geophagus brasiliensis* [22] and *Geophagus aff. Brasiliensis* [31], where anthropogenic impacts were appointed as the causes. These results differ from those described for *Gymnogeophagus setequedas* which did not show recent bottlenecks, possibly for having a large and stable population or for stablishing contact among different lineages [35].

In contrast with the effective population size in PHI (< 1,000), natural environment populations and stocks S4-S5 and S6-S7-S8 exhibited effective population sizes above 1,000, showing a high long-term evolutionary potential [130]. Furthermore, the effective population size in stocks followed a genetic diversity-associated pattern (Ho, He, Ar) concordant with the indirect relationship between the effective population size and the genetic variation observed in other studies [25, 131] that are influenced by maturity age and life expectancy [26]. Studies on effective population sizes in cichlids are relatively limited. Within the examined area, certain low-migration-range species (*Astyanax caucanus*, [132]) and medium-migration-range species (*Prochilodus magdalenae*, [121]) exhibit high effective population sizes. However, *Pseudoplatytoma magdaleniatum*, also classified as a medium-migration-range species, displayed effective population sizes below 1000 [49].

## Conclusions

This study unveils a spatial structure within *C. kraussii*, comprising three genetic groups that exhibit high genetic diversity compared to neotropical cichlids. Both natural stocks, located upstream and downstream the Nechí River mouth, display high effective population size, indicating a high long-term evolutionary potential. Nevertheless, the dam group, probably originated by founder effect, is susceptible to potential harmful effects because of their low effective population size and high relatedness degree. Additionally, this study developed a group of 16 highly polymorphic species-specific microsatellite loci for *C. kraussii* that are proposed as a tool for the future genetic population monitoring of the species. The information obtained reveals the importance of providing the fishing and consumption of the species a differentiated management at local level and contributes to the knowledge of factors modulating the population genetics of neotropical cichlids.

## Supporting information

**S1 Fig. Correlation of genetic diversity and distance to the Cauca River mouth.** Ar: allelic richness; He: expected heterozygosity. He vs Distance to the Cauca River mouth (R: -0.970, p: 0.001). Ar vs Distance to the Cauca River mouth (R: 1.000, p < 0.0001).
(TIF)

**S2 Fig. Mantel test between genetic differentiation and geographical distances.** F'st vs Geographical distance: R = 0.633, p = 0.045. Jost's $D_{EST}$ vs Geographical distance: R = 0.717, p = 0.009.
(TIF)

**S1 Table. Microsatellite loci not selected due to pitfalls in amplification or low levels of polymorphisms.** Ra: Allelic range. Na: number of alleles per locus. PIC: polymorphism information content. Tail A: GCCTCCCTCGCGCCA. Tail B: GCCTTGCCAGCCCGC. Tail C: CAGGACCAGGCTACCGTG. Tail D: CGGAGAGCCGAGAGGTG.
(DOCX)

**S2 Table. Genotype data at 16 microsatellite *loci* included in the population genetic analysis of *Caquetaia kraussii*.** First row indicates respectively: Number of *loci*, Number of individuals, Number of sampling sections, Sample size for seven sections. Third row indicates respectively: Sample ID, Sampling section, Locus name for 16 *loci*.
(XLSX)

**S3 Table. Linkage disequilibrium tests of 16 loci developed for *Caquetaia kraussii*.**
(DOCX)

**S4 Table. Genetic diversity estimators of *Caquetaia kraussii* by site and genetic group in the medium and lower sections of the Cauca River.**
(DOCX)

**S5 Table. Results of the Bayesian probability method for loci detection under selection in *a posteriori* paired comparisons of populations.** PO: posterior probability. Values in bold denote statistical significance.
(DOCX)

**S6 Table. Number and percentage of couples with relatedness in the sections (PHI, S4, S5, S6, S7, and S8) and genetic groups (Stock S4-S5, Stock S6-S7-S8).** PC: Paired comparisons, rxy: Relatedness.
(DOCX)

**S7 Table. Migration rates of *Caquetaia kraussii* among sections (PHI, S4, S5, S6, S7, and S8) of the Cauca River.**
(DOCX)

## Acknowledgments

The authors express their gratitude to Universidad de Antioquia and Universidad de Córdoba for providing the preserved fish tissue samples.

## Author Contributions

**Conceptualization:** Isaí Cataño Tenorio, Cristhian Danilo Joya, Edna Judith Márquez.

**Data curation:** Isaí Cataño Tenorio, Cristhian Danilo Joya, Edna Judith Márquez.

**Formal analysis:** Isaí Cataño Tenorio, Cristhian Danilo Joya, Edna Judith Márquez.

**Funding acquisition:** Edna Judith Márquez.

**Investigation:** Isaí Cataño Tenorio, Edna Judith Márquez.

**Methodology:** Isaí Cataño Tenorio, Cristhian Danilo Joya, Edna Judith Márquez.

**Project administration:** Edna Judith Márquez.

**Resources:** Edna Judith Márquez.

**Supervision:** Edna Judith Márquez.

**Validation:** Isaí Cataño Tenorio, Cristian Danilo Joya, Edna Judith Márquez.

**Visualization:** Isaí Cataño Tenorio, Cristian Danilo Joya, Edna Judith Márquez.

**Writing – original draft:** Isaí Cataño Tenorio, Cristian Danilo Joya, Edna Judith Márquez.

**Writing – review & editing:** Isaí Cataño Tenorio, Cristian Danilo Joya, Edna Judith Márquez.

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
