## [Decision Letter · Decision Letter 0]

23 Jan 2024

PONE-D-23-37840Spatial population genetic structure of Caquetaia kraussii (Steindachner, 1878) evidenced by species-specific microsatellite loci in the middle and low basin of the Cauca River, ColombiaPLOS ONE

Dear Dr. Marquez,

Thank you for submitting your manuscript to PLOS ONE. After careful consideration, we feel that it has merit but does not fully meet PLOS ONE’s publication criteria as it currently stands. Therefore, we invite you to submit a revised version of the manuscript that addresses the points raised during the review process.

Both reviewer's have some valid critisizm of the analyses and the manuscript that need to be addressed before acceptance. Please revise the manuscript accordingly and carefully consider the comments regarding the soundness of the analyses. 

We look forward to receiving your revised manuscript.

Kind regards,

Sven Winter

Academic Editor

PLOS ONE

Journal Requirements:

"This study was funded by Universidad Nacional de Colombia Sede Medellín and Empresas Públicas de Medellín, Grant CT-2019-000661 “Variabilidad genética de un banco de peces de los sectores medio y bajo del Río Cauca”."

"This study was funded by Universidad Nacional de Colombia Sede Medellín and Empresas Públicas de Medellín, Grant CT-2019-000661 “Variabilidad genética de un banco de peces de los sectores medio y bajo del Río Cauca”. Funders have not play any role in the study design, data collection and analysis, decision to publish, or preparation of the manuscript."

6. We note that [Figure 1] in your submission contain [map/satellite] images which may be copyrighted. All PLOS content is published under the Creative Commons Attribution License (CC BY 4.0), which means that the manuscript, images, and Supporting Information files will be freely available online, and any third party is permitted to access, download, copy, distribute, and use these materials in any way, even commercially, with proper attribution. For these reasons, we cannot publish previously copyrighted maps or satellite images created using proprietary data, such as Google software (Google Maps, Street View, and Earth). For more information, see our copyright guidelines: http://journals.plos.org/plosone/s/licenses-and-copyright.

Reviewers' comments:

Reviewer's Responses to Questions

**Comments to the Author**

1. Is the manuscript technically sound, and do the data support the conclusions?

Reviewer #1: Partly

Reviewer #2: Yes

2. Has the statistical analysis been performed appropriately and rigorously? 

Reviewer #1: No

Reviewer #2: Yes

3. Have the authors made all data underlying the findings in their manuscript fully available?

Reviewer #1: No

Reviewer #2: Yes

4. Is the manuscript presented in an intelligible fashion and written in standard English?

Reviewer #1: Yes

Reviewer #2: Yes

5. Review Comments to the Author

Reviewer #1: Comments to: “Spatial population genetic structure of Caquetaia kraussii (Steindachner, 1878) evidenced by species-specific microsatellite loci in the middle and low basin of the Cauca River, Colombia” by IC Tenorio, CD Joya and EJ Márquez.

This MS describe population genetic structure and genetic diversity of a cichlid species. Using variability of 16 microsatellite loci the authors found high genetic diversity and four genetic clusters.

The MS is well written and easy to follow. I have comments.

1.- The manuscript is not in the format of the journal. Authors should consider the submission instructions for their papers.

2.- MS lacks page number and line numbers

Introduction

3.- Page 1 introduction. Please add a reference to: “Moreover, individuals that migrate and reproduce with distant populations exhibit gene flow and, thus, high genetic diversity.”

4.- “Particularly, this study aimed to provide a response to C. kraussii genetic diversity and

structure related questions between 2020 and 2022”. I cannot find where the authors used date (2020-2022) in the analysis. Please explain or delete.

Materials and methods

5.- “Furthermore, a discriminant analysis of principal components (DAPC) was

performed using the R-package Adegenet (Jombart, 2008), with six principal components

retained.” How the authors selected 6 PCA? How many DL were used? The authors used de a-score?

6.- “Other approach included a Bayesian analysis in STRUCTURE 2.3.4 (Pritchard et al., 2000), with 1,000,000 Monte Carlo Markov chains with 100.000 regarded as the burn-in period.” The authors used the admixture model? The authors used correlated or uncorrelated allele frequencies?

7.- “BayeScan v2.1 (Foll & Gaggiotti, 2008) was used for loci detection under selection.” BayeScan could be used before all analysis.

8.- “Lastly, GENECLASS2 (Piry et al., 2004) was used for testing whether an individual resides in the sampled site or is immigrant from another section. To this end, L_home/L_max rate was used for first generation migrant detection.” Why you expect first generation migrant? Please use other methods (e.g. BayesAss).

Results:

9.- “Genetic diversity showed a decreasing gradient from the lower to the middle section of the Cauca River (S8>S7>S6>S5>S4)” Please add a figure for this result.

10.- “for both He (R: -0.970, p: 0.001) and Ar (R: 1.000, p: 0.000)”. Please, never a p-value = 0. Probability in this case could be p < 0.0001.

11.- Table 2.: please careful with p-value. Change from 0.0000 to <0.0001.

12.- Table 2: Change from “PHWE: statistical significance tests of departure from Hardy-Weinberg equilibrium.” To PHWE: p-value for the departure from Hardy-Weinberg equilibrium. Same for FIS p-value

13.- “Furthermore, Bayescan posterior probability (PO) values for the loci detection under selection (Supplementary Table 4) evidenced that the only paired comparison showing one locus under selection (Ckra 21; reference value > 0.76) was S4-S5 v. S6-S7-S8. Moreover, another parameter found, Alpha, had a value of 1.234 which suggests the existence of balancing or purifying selection. It is important in the analysis this detection? Do you erase these loci? No words in discussion about that?

14.- “(Supplementary Figure 2)”: I cannot find this figure.

Discussion

Please re-write this section considering a whole discussion follow the main goal of the study. Discussion about each method used cannot be useful for readers. For example, main results are about population genetic structure. Thus, what about the respective role of the barriers and fish behavior, etc.

Discussion about Ne is poor. Please improve this section comparing with other species.

Conclusion

Please re-write considering main results.

References

Please change “Referencias” to “References”

Change format to plosone

Figures

Change quality (300 dpi)

Reviewer #2: The manuscript contains an interesting study on the genetic diversity and populations structure of the cichlid Caquetaia Krausii located in the basin of the Causa River. The topic is worthy of investigation, and the manuscript is particularly interesting in that it investigates the effect of Ituango hydroelectric project on the genetic structure of the species.

In principle, the experimental design is appropriate, except that the study of EPSC stock does not fit in. The objective of the manuscript is to determine the genetic diversity of the species and the genetic structure of the population. In my opinion, the main factor in examining the genetic structure is the impact of the hydroelectric project on the natural populations, and the farm stock does not correspond to the natural stock.

The structure of the introduction is appropriate; however, I suggest including a paragraph on the hydroelectric project and the effect of such projects on the genetic diversity of cichlids.

The description of the methodology needs some corrections:

Please add sample sizes in the materials and methods. They could be found only in Table 2 in the results.

The 16 microsatellite markers were selected from 30 “according to their polymorphism level” – this means that 14 were monomorphic, or that the PIC value was not enough high?

Please add the description of the Mantel test to the materials and methods.

Please add the proportion of SMM in TPM, the variance of TPM and the number of iterations for the Wilcoxon signed rank test.

Results:

The authors give a negative correlation between Ar and geographical distance, but the value of R is 1.000 (based on the figure, this does not seem realistic). Please revise it.

In Table 2, are the values means of the 16 markers? If yes, then please include SD also. Was there a statistical comparison between populations? If yes, significant differences should be marked and taken into account in the diversity gradient in the text.

The resolution of Figure 2 is low, making the figure difficult to interpret.

Additionally to the first-generation migrant detection, I suggest calculating the relative directional migration network by the method based on Jost's D using divMigrate-online software (Sundqvist et al., 2016). In addition to a statistical assessment of one-way migration, this also provides a good and easy-to-understand chart to illustrate the extent of migration.

I recommend running the mantel test without the farm stock, as it is expected to improve the correlation. Moreover, it might be also useful to calculate only the stocks under the hydroelectric project separately, taking into account its strong impact.

Discussions should focus less on the reproductive features of the species and more on the habitat characteristics that may cause differences in population size and diversity of subpopulations. This is particularly valid for the hydroelectric project (founder effect, relatedness, migration rate, etc.).

Nevertheless, after the necessary corrections, the manuscript could provide valuable results for the population genetics of Cichlid fishes.

6. PLOS authors have the option to publish the peer review history of their article (what does this mean?). If published, this will include your full peer review and any attached files.

Reviewer #1: No

Reviewer #2: No

---

## [Author Response · Author response to Decision Letter 0]

11 Mar 2024

Response to reviewers

Dear Editor,

We really appreciate the detailed revision of the reviewers and have edited the manuscript following your valuable recommendations, which have led to improve our paper.

We hope that the manuscript is now suitable for publication in Plos One.

PONE-D-23-37840

Spatial population genetic structure of Caquetaia kraussii (Steindachner, 1878) evidenced by species-specific microsatellite loci in the middle and low basin of the Cauca River, Colombia

PLOS ONE

Dear Dr. Marquez,

Thank you for submitting your manuscript to PLOS ONE. After careful consideration, we feel that it has merit but does not fully meet PLOS ONE’s publication criteria as it currently stands. Therefore, we invite you to submit a revised version of the manuscript that addresses the points raised during the review process.

Both reviewer's have some valid critisizm of the analyses and the manuscript that need to be addressed before acceptance. Please revise the manuscript accordingly and carefully consider the comments regarding the soundness of the analyses. 

Please see below.

Complete information is documented in the manuscript.

We look forward to receiving your revised manuscript.

Kind regards,

Sven Winter

Academic Editor

PLOS ONE

Journal Requirements:

Done. Now: “This study assessed fin and muscle tissues preserved in ethanol 95%, obtained from 403 C. kraussii individuals captured in different sectors of the Cauca River (Figure 1) by Universidad de Antioquia, Universidad de Córdoba, and Universidad Nacional de Colombia Sede Medellín, through scientific cooperation agreement CT-2019-000661, under environmental license # 0155 of January 30th, 2009, from Ministry of Environment, Housing and Territorial Development for the Ituango hydroelectric construction.”

Done. Now we include “S2 Table. Genotype data at 16 microsatellite loci included in the population genetic analysis of Caquetaia kraussi. First row indicates respectively: Number of loci, Number of individuals, Number of sampling sections, Sample size fo seven sections. Third row indicates respectively: Sample ID, Sampling section, Genetic stock, Locus name for 16 loci.”

"This study was funded by Universidad Nacional de Colombia Sede Medellín and Empresas Públicas de Medellín, Grant CT-2019-000661 “Variabilidad genética de un banco de peces de los sectores medio y bajo del Río Cauca”."

"This study was funded by Universidad Nacional de Colombia Sede Medellín and Empresas Públicas de Medellín, Grant CT-2019-000661 “Variabilidad genética de un banco de peces de los sectores medio y bajo del Río Cauca”. Funders have not play any role in the study design, data collection and analysis, decision to publish, or preparation of the manuscript."

Done. We have removed the text, and we would like to retain the current Funding Statement.

Please see response above.

6. We note that [Figure 1] in your submission contain [map/satellite] images which may be copyrighted. All PLOS content is published under the Creative Commons Attribution License (CC BY 4.0), which means that the manuscript, images, and Supporting Information files will be freely available online, and any third party is permitted to access, download, copy, distribute, and use these materials in any way, even commercially, with proper attribution. For these reasons, we cannot publish previously copyrighted maps or satellite images created using proprietary data, such as Google software (Google Maps, Street View, and Earth). For more information, see our copyright guidelines: http://journals.plos.org/plosone/s/licenses-and-copyright.

Now, we provide a new figure 1. “Figure 1. Caquetaia kraussii sampled sites in the Cauca River, Colombia.

PHI: Ituango hydroelectric project, 17 sites (purple). ECrPSC1: Santa Cruz fish farming station, 1 site (gray). S4: 3 sites (light blue). S5: 5 sites (blue). S6: 13 sites (red). S7: 3 sites (orange). S8: 2 sites (yellow). Self-made creation of the map based on contour lines scaled 1:100,000 from the Instituto Geográfico Agustín Codazzi source, 2019 (Available from: IGAC Geoportal, https://geoportal.igac.gov.co/contenido/datos-abiertos-cartografia-y-geografia).”

Done. Legends of the Supporting Information files are now at the end of the manuscript.

Reviewers' comments:

Reviewer's Responses to Questions

Comments to the Author

1. Is the manuscript technically sound, and do the data support the conclusions?

Reviewer #1: Partly

Reviewer #2: Yes

2. Has the statistical analysis been performed appropriately and rigorously?

Reviewer #1: No

Reviewer #2: Yes

3. Have the authors made all data underlying the findings in their manuscript fully available?

Reviewer #1: No

Reviewer #2: Yes

4. Is the manuscript presented in an intelligible fashion and written in standard English?

Reviewer #1: Yes

Reviewer #2: Yes

5. Review Comments to the Author

Reviewer #1: Comments to: “Spatial population genetic structure of Caquetaia kraussii (Steindachner, 1878) evidenced by species-specific microsatellite loci in the middle and low basin of the Cauca River, Colombia” by IC Tenorio, CD Joya and EJ Márquez.

This MS describe population genetic structure and genetic diversity of a cichlid species. Using variability of 16 microsatellite loci the authors found high genetic diversity and four genetic clusters.

The MS is well written and easy to follow. I have comments.

1.- The manuscript is not in the format of the journal. Authors should consider the submission instructions for their papers.

2.- MS lacks page number and line numbers

Done

Introduction

3.- Page 1 introduction. Please add a reference to: “Moreover, individuals that migrate and reproduce with distant populations exhibit gene flow and, thus, high genetic diversity.”

Done. Now: “Moreover, individuals that migrate and reproduce with distant populations exhibit gene flow and, thus, high genetic diversity [24,25].”

4.- “Particularly, this study aimed to provide a response to C. kraussii genetic diversity and

structure related questions between 2020 and 2022”. I cannot find where the authors used date (2020-2022) in the analysis. Please explain or delete.

Done. Now: “Namely, this study aimed to provide a response to C. kraussii genetic diversity and structure related questions in the medium and lower sections of the Cauca River”

Materials and methods

5.- “Furthermore, a discriminant analysis of principal components (DAPC) was

performed using the R-package Adegenet (Jombart, 2008), with six principal components

retained.” How the authors selected 6 PCA? How many DL were used? The authors used de a-score?

This idea is incomplete. Now: “Furthermore, a discriminant analysis of principal components (DAPC) was performed using the R-package Adegenet (Jombart, 2008), with 32 principal components and six discriminant functions retained.”

a-score: 0.422.

6.- “Other approach included a Bayesian analysis in STRUCTURE 2.3.4 (Pritchard et al., 2000), with 1,000,000 Monte Carlo Markov chains with 100.000 regarded as the burn-in period.” The authors used the admixture model? The authors used correlated or uncorrelated allele frequencies?

Done. Now: “Other approach included a Bayesian analysis in STRUCTURE 2.3.4 [94], with 1,000,000 Monte Carlo Markov chains with 100,000 regarded as the burn-in period, admixture model and correlated allele frequencies as a priori models.”

7.- “BayeScan v2.1 (Foll & Gaggiotti, 2008) was used for loci detection under selection.” BayeScan could be used before all analysis.

BayeScan was run prior the other analysis. W

---

## [Decision Letter · Decision Letter 1]

12 Apr 2024

PONE-D-23-37840R1Spatial population genetic structure of Caquetaia kraussii (Steindachner, 1878) evidenced by species-specific microsatellite loci in the middle and low basin of the Cauca River, ColombiaPLOS ONE

Dear Dr. Márquez,

Thank you for submitting your revised manuscript to PLOS ONE. After careful consideration, we feel that it has merit but does not fully meet PLOS ONE’s publication criteria as it currently stands. Therefore, we invite you to submit a revised version of the manuscript that addresses the points raised during the review process.

We look forward to receiving your revised manuscript.

Kind regards,

Sven Winter

Academic Editor

PLOS ONE

Journal Requirements:

**Additional Editor Comments:**

The manuscript has greatly improved, and the reviewers were mostly satisfied with the changes. Before acceptance, however, I would like a more detailed reply to the open question regarding the inclusion of the EPSC stock, as this was repeatedly criticized by reviewer 2. 

Reviewers' comments:

Reviewer's Responses to Questions

**Comments to the Author**

1. If the authors have adequately addressed your comments raised in a previous round of review and you feel that this manuscript is now acceptable for publication, you may indicate that here to bypass the “Comments to the Author” section, enter your conflict of interest statement in the “Confidential to Editor” section, and submit your "Accept" recommendation.

Reviewer #1: All comments have been addressed

Reviewer #2: (No Response)

2. Is the manuscript technically sound, and do the data support the conclusions?

Reviewer #1: Yes

Reviewer #2: Yes

3. Has the statistical analysis been performed appropriately and rigorously? 

Reviewer #1: Yes

Reviewer #2: Yes

4. Have the authors made all data underlying the findings in their manuscript fully available?

Reviewer #1: No

Reviewer #2: Yes

5. Is the manuscript presented in an intelligible fashion and written in standard English?

Reviewer #1: Yes

Reviewer #2: Yes

6. Review Comments to the Author

Reviewer #1: the authors answered all my questions. the authors performed new analyses, corrected typos and rewrote the discussion

Reviewer #2: The authors have made significant changes to the manuscript, taking into account most of my suggestions. I accept the methodology and justification for the migration estimation. However, I still see no justification for including the EPSC stock in the analysis, as it is a farm stock, not a natural one. As a basis for comparison (as explained by the authors), I can only accept it if the genetic characteristics of the PHI (above dam) stock are compared with those of the EPSC stock, since it was assumed that isolation is observed for both stocks. However, this would require more data on the EPSC stock (how long it has been bred in isolation, what founder stock size it started with, etc.), and would then give a good indication of what the dam construction has caused in the PHI stock in comparison. This was partly done in the discussion, it is worth highlighting the changes in more detail.

In addition, Figure 3 showing the STRUCTURE analysis results is missing from the uploaded manuscript.

Nevertheless, with the above suggestions, I consider the manuscript publishable.

7. PLOS authors have the option to publish the peer review history of their article (what does this mean?). If published, this will include your full peer review and any attached files.

Reviewer #1: No

Reviewer #2: No

---

## [Author Response · Author response to Decision Letter 1]

14 May 2024

Response to reviewers

Dear Editor,

We really appreciate the detailed revision of the reviewers and have edited the manuscript following your valuable recommendations, which have led to improve our paper.

We hope that the manuscript is now suitable for publication in Plos One.

PONE-D-23-37840R1

Spatial population genetic structure of Caquetaia kraussii (Steindachner, 1878) evidenced by species-specific microsatellite loci in the middle and low basin of the Cauca River, Colombia

PLOS ONE

Dear Dr. Márquez,

Thank you for submitting your revised manuscript to PLOS ONE. After careful consideration, we feel that it has merit but does not fully meet PLOS ONE’s publication criteria as it currently stands. Therefore, we invite you to submit a revised version of the manuscript that addresses the points raised during the review process.

We look forward to receiving your revised manuscript.

Kind regards,

Sven Winter

Academic Editor

PLOS ONE

Journal Requirements:

Additional Editor Comments:

The manuscript has greatly improved, and the reviewers were mostly satisfied with the changes. Before acceptance, however, I would like a more detailed reply to the open question regarding the inclusion of the EPSC stock, as this was repeatedly criticized by reviewer 2. 

Reviewers' comments:

Reviewer's Responses to Questions

Comments to the Author

1. If the authors have adequately addressed your comments raised in a previous round of review and you feel that this manuscript is now acceptable for publication, you may indicate that here to bypass the “Comments to the Author” section, enter your conflict of interest statement in the “Confidential to Editor” section, and submit your "Accept" recommendation.

Reviewer #1: All comments have been addressed

Reviewer #2: (No Response)

2. Is the manuscript technically sound, and do the data support the conclusions?

Reviewer #1: Yes

Reviewer #2: Yes

3. Has the statistical analysis been performed appropriately and rigorously?

Reviewer #1: Yes

Reviewer #2: Yes

4. Have the authors made all data underlying the findings in their manuscript fully available?

Reviewer #1: No

In the first review, we included “S2 Table. Genotype data at 16 microsatellite loci included in the population genetic analysis of Caquetaia kraussi. First row indicates respectively: Number of loci, Number of individuals, Number of sampling sections, Sample size fo seven sections. Third row indicates respectively: Sample ID, Sampling section, Genetic stock, Locus name for 16 loci.”

Reviewer #2: Yes

5. Is the manuscript presented in an intelligible fashion and written in standard English?

Reviewer #1: Yes

Reviewer #2: Yes

6. Review Comments to the Author

Reviewer #1: the authors answered all my questions. the authors performed new analyses, corrected typos and rewrote the discussion

Reviewer #2: The authors have made significant changes to the manuscript, taking into account most of my suggestions. I accept the methodology and justification for the migration estimation. However, I still see no justification for including the EPSC stock in the analysis, as it is a farm stock, not a natural one. As a basis for comparison (as explained by the authors), I can only accept it if the genetic characteristics of the PHI (above dam) stock are compared with those of the EPSC stock, since it was assumed that isolation is observed for both stocks. However, this would require more data on the EPSC stock (how long it has been bred in isolation, what founder stock size it started with, etc.), and would then give a good indication of what the dam construction has caused in the PHI stock in comparison. This was partly done in the discussion, it is worth highlighting the changes in more detail.

In addition, Figure 3 showing the STRUCTURE analysis results is missing from the uploaded manuscript.

Nevertheless, with the above suggestions, I consider the manuscript publishable.

Following the original suggestion of the reviewer, we have removed the EPSC stock from all sections of the current version.

7. PLOS authors have the option to publish the peer review history of their article (what does this mean?). If published, this will include your full peer review and any attached files.

Do you want your identity to be public for this peer review? For information about this choice, including consent withdrawal, please see our Privacy Policy.

Reviewer #1: No

Reviewer #2: No

---

## [Editor Report · Decision Letter 2]

20 May 2024

Spatial population genetic structure of Caquetaia kraussii (Steindachner, 1878) evidenced by species-specific microsatellite loci in the middle and low basin of the Cauca River, Colombia

PONE-D-23-37840R2

Dear Dr. Márquez,

We’re pleased to inform you that your manuscript has been judged scientifically suitable for publication and will be formally accepted for publication once it meets all outstanding technical requirements.

Kind regards,

Sven Winter

Academic Editor

PLOS ONE
---

## [Editor Report · Acceptance letter]

24 May 2024

PONE-D-23-37840R2 

PLOS ONE

Dear Dr. Márquez, 

I'm pleased to inform you that your manuscript has been deemed suitable for publication in PLOS ONE. Congratulations! Your manuscript is now being handed over to our production team.

Kind regards, 

on behalf of

Dr. Sven Winter 

Academic Editor

PLOS ONE